# Studies on Silver Ions Releasing Processes and Mechanical Properties of Surface-Modified Titanium Alloy Implants

**DOI:** 10.3390/ijms19123962

**Published:** 2018-12-09

**Authors:** Aleksandra Radtke, Marlena Grodzicka, Michalina Ehlert, Tadeusz M. Muzioł, Marek Szkodo, Michał Bartmański, Piotr Piszczek

**Affiliations:** 1Faculty of Chemistry, Nicolaus Copernicus University in Toruń, Gagarina 7, 87-100 Toruń, Poland; aradtke@umk.pl (A.R.); marlena.grodzicka@doktorant.umk.pl (M.G.); m.ehlert@doktorant.umk.pl (M.E.); tmuziol@chem.umk.pl (T.M.M.); 2Nano-Implant Ltd. Jurija Gagarina 5/102, 87-100 Toruń, Poland; 3Faculty of Mechanical Engineering, Gdańsk University of Technology, ul. Gabriela Narutowicza 11/12, 80-233 Gdańsk, Poland; marek.szkodo@pg.edu.pl (M.S.); michal.bartmanski@pg.edu.pl (M.B.)

**Keywords:** titanium alloy, silver nanoparticles, surface morphology, mechanical properties, surface free energy, silver ions release

## Abstract

Dispersed silver nanoparticles (AgNPs) on the surface of titanium alloy (Ti6Al4V) and titanium alloy modified by titania nanotube layer (Ti6Al4V/TNT) substrates were produced by the chemical vapor deposition method (CVD) using a novel precursor of the formula [Ag_5_(O_2_CC_2_F_5_)_5_(H_2_O)_3_]. The structure and volatile properties of this compound were determined using single crystal X-ray diffractometry, variable temperature IR spectrophotometry (VT IR), and electron inducted mass spectrometry (EI MS). The morphology and the structure of the produced Ti6Al4V/AgNPs and Ti6Al4V/TNT/AgNPs composites were characterized by scanning electron microscopy (SEM) and atomic force microscopy (AFM). Moreover, measurements of hardness, Young’s modulus, adhesion, wettability, and surface free energy have been carried out. The ability to release silver ions from the surface of produced nanocomposite materials immersed in phosphate-buffered saline (PBS) solution has been estimated using inductively coupled plasma mass spectrometry (ICP-MS). The results of our studies proved the usefulness of the CVD method to enrich of the Ti6Al4V/TNT system with silver nanoparticles. Among the studied surface-modified titanium alloy implants, the better nano-mechanical properties were noticed for the Ti6Al4V/TNT/AgNPs composite in comparison to systems non-enriched by AgNPs. The location of silver nanoparticles inside of titania nanotubes caused their lowest release rate, which may indicate suitable properties on the above-mentioned type of the composite for the construction of implants with a long term antimicrobial activity.

## 1. Introduction

The design and the manufacture of customized implants using innovative technologies is one of the main directions in modern implantology development [1,2]. New generation implants fabrication besides their anatomic fit [3,4] requires the development of new alloys and composite coatings, which provide them suitable biointegration properties, but also improve their mechanical properties, durability, hydrophilicity, etc. The implant, in order to be effective, must not only restore the function of the organ, but also ensure the patient’s comfort, and protect him from negative effects, e.g., formation of inflammation or allergic reactions. The customization of implants in the patient anatomy is associated with the development of the numeric image techniques and such three-dimensional (3D) printing technology as selective laser sintering (SLS) and selective laser melting (SLM). Both above-mentioned techniques allow for the formation of three-dimensional metal structures by selective melting of metal powder in a layer-by-layer manner, which enable the formation of products with new chemical properties, differing from their macroscopic equivalents [5,6,7]. The response of the tissues to the implant is largely controlled by the implant surface morphology and properties. When compared to smooth surfaces, textured implants surfaces exhibit larger surface area for integrating with bone, via osseointegration process. Improved bone bonding and accelerated bone formation appears to be possible with roughened surfaces that are modified with certain treatments, which can be classified into mechanical, physical, chemical, and electrochemical methods [8]. Our earlier works on the modification of titanium and Ti6Al4V implants have shown that the fabrication of TiO_2_ nanotube (TNT) layers of strictly defined tubes diameter on their surface had an impact on the cell adhesion and proliferation improvement [9,10].

Another problem, which should be taken into account during the designing new generation of implants, is the appearance of complications after implant introduction—a possible bacterial infections. Infections that are related to foreign body are difficult to treat because the bacteria, which cause these infections, live in well-developed biofilms. In this way, there are protected against the action of antimicrobials [11,12]. The providing the antimicrobial activity of implant surface is a complicated issue due to the necessity of the antimicrobial coatings use, which should be universal versus different strains of bacteria that are present in the organism and/or introduced with the implant [13]. Surface-modified implants with a layer of titanium dioxide can be enriched with biocides: antibiotics or other antibacterial agents. Antibiotics can be used for this purpose, however, due to bacterial resistance and concerns about their long-term effectiveness, they may not produce the desired effects [14]. Silver is the most well known antimicrobial agent of low toxicity to mammalian cells and it is effective against more than 650 pathogens. According results of previous studies, it should be noted that AgNPs are one of the most viable alternatives to antibiotics and they may be used in a wide range of applications [15,16,17,18,19,20]. AgNPs can be synthesized using the sol-gel method, electrophoretic deposition from aqueous suspensions, physical vapor deposition (PVD), chemical vapor deposition (CVD), and atomic layer deposition (ALD) [21,22,23,24]. In our works, we have focused on the use of CVD methods, which allow the formation of dispersed AgNPs on the substrate surface. The shape, size, and dispersion level of silver nanoparticles can be controlled, by optimizing the deposition parameters and also by the type of chemical compound that is used as a precursor [25]. Silver(I) complexes with fluorinated or non-fluorinated β-diketonates/carboxylates and tertiary phosphines are the most commonly used as precursors in these techniques [26,27]. Also, the selected silver(I) carboxylates can be applied as a solid source of metallic particles in CVD techniques, within silver(I) pentafluoropropionate (Ag(O_2_CC_2_F_5_), as an example. The advantage of this compound is suitable volatility, low decomposition temperature at low vacuum, and a short deposition time. Moreover, it is characterized by simple and cheap synthesis [28]. In this paper, we present the results of the use of trihydrate of the above-mentioned compound as a new silver CVD precursor. The carried out studies concern the optimization of a CVD method for the production of dispersed AgNPs on the surface of Ti6Al4V implants that were manufactured by the SLS method, as obtained and modified by titanium dioxide nanotubes of different diameters. The important part of our works was the estimation of wettability, surface roughness, and mechanical properties of the produced implants. The results concerning the above-mentioned issues are not widely discussed in previous reports. Moreover, the silver ions releasing from the surface is discussed in the presented paper. It is especially important for the potential application of Ti6Al4V/AgNPs and T6Al4V/TNT/AgNPs composite materials in the construction of customized implants.

## 2. Results

### 2.1. The Chemical Vapor Deposition of Silver Nanoparticles

#### 2.1.1. Precursor—The Structure and Thermal Properties of [Ag_5_(O_2_CC_2_F_5_)_5_(H_2_O)_3_]

Simple and inexpensive synthesis of silver(I) pentafluoropropionate, and the especially suitable properties of this compound as a silver CVD precursor decided its choice for all of our experiments related to the enrichment of Ti6Al4V implants by AgNPs [28,29]. The purification of Ag(O_2_CC_2_F_5_) by the slow recrystallization of this compound from anhydrous ethanol enabled obtaining the needle-like crystals, which quality did not allow for determining their structure on the base of single crystals x-ray diffractometry. The use of 1:2 EtOH/H_2_O mixture in the recrystallization process allowed for the isolation of colorless crystals after five days. Their stability in air and light was higher than pure Ag(O_2_CC_2_F_5_). Analysis of single crystal X-ray diffraction data exhibits the formation of Ag(I) complex of the formula [Ag_5_(O_2_CC_2_F_5_)_5_(H_2_O)_3_], which crystallizes in the triclinic system, space group P-1 (Figure 1, Table 1).

The structure of this complex is formed by five differently surrounded Ag(I) atoms, which are linked by carboxylate bridges and water molecules. However, the presence of three water molecules (O_w_ = O7, O8, O9) in the structure of this Ag(I) compound influences on its novelty and thereby its use as a new silver CVD precursor. Analysis of the single crystal X-ray diffraction data revealed that O7 bridges Ag3 and Ag4 atoms, forming the metal-oxide bonds of lengths 2.547 and 2.426 Å (Table 1). Simultaneously, O12-O7 and O22-O7 of 2.776 and 2.729 Å suggest the formation of middle hydrogen bonds [30]. Whereas, O8 and O9 molecules are strongly bonded by Ag4 (2.577 Å) and Ag6 (2.318 Å) and they are in the field of interactions with Ag1 (2.825 Å) and Ag5 (2.545 Å) (Table 1). The O_w_-O and O_w_-F distances below 2.8–3.4 Å, which were found between O42-O8-F70 (2.924 and 3.404 Å), O11-O9-F15 (2.777 and 3.051 Å) suggest the formation of middle and weak hydrogen bonds.

Results of the thermal analysis (TG/DTG/DTA) revealed that the thermal decomposition of this compound proceeds in one step and it is an endothermic process (beginning—453 K, max—598 K, and ending—613 K).

The analysis of the TG curve revealed that during heating of [Ag_5_(O_2_CC_2_F_5_)_5_(H_2_O)_3_] between 298 and 773 K under an inert atmosphere (N_2_), the weight loss of the studied sample was c.a. 63.3%. It suggested that the metallic silver was a final product of this compound pyrolysis. To assess the volatility of the isolated Ag(I) compound, the variable temperature IR (VTIR) and the mass spectrometry (MS EI) studies have been carried out. The use of the VTIR method allowed for the estimation of the thermal stability of isolated crystals in the temperature range 303-523 K. According to VTIR data, the dehydration of [Ag_5_(O_2_CC_2_F_5_)_5_(H_2_O)_3_] (disappearance of bands at 3436 and 3669 cm^−1^) and the clear changes in the way of carboxyl groups interaction with Ag(I) ions (splitting of the ν_as_(COO) band) were found between 303 and 398 K (Figure 2). The further heating of the studied compound of up to 523 K led to the formation of the stable system, which consisted of dimeric species. It was confirmed by the appearance of a single ν_as_(COO) band at 1690 cm^−1^ [28].

The use of mass spectrometry (MS EI) allowed for the determination of the vapor composition at temperatures 403 and 513 K during the heating of [Ag_5_(O_2_CC_2_F_5_)_5_(H_2_O)_3_] (Table 2) [28,29]. Analysis of these data allowed for identifying the following silver(I) containing species: [Ag(O_2_CC_2_F_5_)(H_2_O)]^+^_,_ [Ag(O_2_CC_2_F_5_)_2_(H_2_O)]^+^, [Ag_2_(O_2_CC_2_F_5_)_3_(OC)(H_2_O)]^+^, and [Ag_2_(O_2_CC_2_F_5_)_3_(OOC) (H_2_O)_2_]^+^ in the spectrum registered at 403 K. It can indicate that dehydration process proceeds with the partial decomposition of trihydrate Ag(I) compound. The data presented in Table 2 suggest that the complete decomposition of this compound proceeds above 503 K, and the following Ag(I) containing species will be transported in vapors: [Ag(C_2_F_5_)]^+^, [Ag_2_(C_2_F_5_)]^+^, and [Ag_2_(O_2_CC_2_F_5_)]^+^. Their appearance in vapors suggests that they can be the main source of the metallic silver in CVD processes.

#### 2.1.2. The Enrichment of Ti6Al4V and Ti6Al4V/TNT Substrates by Silver Nanoparticles (AgNPs)

When considering the results of thermal decomposition and volatility studies of [Ag_5_(O_2_CC_2_F_5_)_5_(H_2_O)_3_], the overall conditions for carrying out the CVD processes were established. The optimal conditions have been determined during deposition experiments and the obtained results are listed in Table 3. The use of scanning electron microscopy (Scanning Electron Microscopy with Energy Dispersive Spectroscopy (SEM/EDS) method allowed for confirming that the result of the deposition processes were metallic silver nanoparticles (Figure 3).

Analysis of SEM images of the Ti6Al4V implant (Figure 4a) that were enriched by AgNPs revealed that the substrate surface is uniformly covered by Ag spherical grains of diameters from 15 up to 27 nm (*r*_D_ = 2.57 mg·min^−1^; Figure 4b). Ti6Al4V/TNT/AgNPs nanocomposites were produced during the two-step procedure. In the first step, the surface of Ti6Al4V implants (produced by the selective laser sintering (SLS) technique) was modified by titania nanotubes layer using the electrochemical anodization method. The anodization process was carried out using 5, 15, and 20V potentials, and the obtained samples were designated as Ti6Al4V/TNT5, Ti6Al4V/TNT15, and Ti6Al4V/TNT20, respectively. The SEM images of the produced Ti6Al4V/TNT nanocomposites are presented in Figure 4c,e,g. According to these data, the produced TNT layers consisted of nanotubes of diameters ca. 35–45 nm (TNT5), 70–80 nm (TNT15), and 100–120 nm (TNT20). Analysis of Raman and DRIFT (diffuse reflectance infrared Furrier transformation) spectra proved the formation TiO_2_ amorphous layers.

Enrichment of TNT layers by AgNPs using CVD technique was the next step. SEM images of modified titanium alloy implants are presented in Figure 4d,f,h. Dependently to the type of the TNT morphology layer, the differences in the size and distribution of AgNPs were noticed. Mass differences before and after CVD process of Ti6Al4V/TNT/AgNPs samples suggest the formation of coatings containing ca. 1.7 wt% (TNT5), 1.4 wt% (TNT15), and 1.9 wt% (TNT20) of silver grains. On the surface of Ti6Al4V/TNT5 coating, the dispersed nanoparticles of diameters 34–80 nm, were localized mainly on the layer surface (*r*_D_ = 2.54 mg·min^−1^; Figure 4d). In the case of TNT15 coating, which consists of tubes of diameters 70–80 nm (TNT15) the size of AgNPs decreased up to 10–18 nm (*r*_D_ = 2.25 mg·min^−1^; Figure 4f). The deposited metallic grains were localized inside of tubes on their walls. A further increase in the nanotubes diameter (TNT20) was accompanied by an increase of the nanograins size up to 25–35 nm (*r*_D_ = 2.42 mg·min^−1^, Figure 4h). Also, in this case, AgNP were located on the surface of tube walls.

### 2.2. Measurement of the Contact Angle and Surface Free Energy of Biomaterials

Wettability of studied coatings surface and their surface free energy (SFE) were estimated using two different liquids, i.e., water as a polar liquid and diiodomethane as a dispersive one. In all studied cases, the values of contact angles, which were measured for water, were larger in comparison to adequate value for diiodomethane (Table 4). According to these data, the wettability of the Ti6Al4V/TNT surfaces increases in the row TNT5 < TNT15 < TNT20 (for TNT layers produced using potential of 5, 15, and 20V, respectively) and is better than for pure Ti6Al4V. The enrichment of these materials with AgNPs leads to the wettability decrease (increase of hydrophobic properties) and surface free energy decrease. 

### 2.3. Mechanical Properties of Ti6Al4V/AgNPs, Ti6Al4V/TNT, and Ti6Al4V/TNT/AgNPs

The studies have been carried out for Ti6Al4V/AgNPs, Ti6Al4V/TNT, and Ti6Al4V/TNT/AgNPs systems, where TNT layers were produced using 5V (TNT5) and 15V (TNT15) potentials. The aim was to estimate mechanical property changes of two different types of TNT coatings that were enriched by AgNPs, i.e., the network formed by densely packed TiO_2_ tubes (TNT5) and the layer composed of separated and ordered nanotubes (TNT15).

#### 2.3.1. Surface Topography

Surface topographies of the obtained coatings and the reference Ti alloy were examined by means of atomic force microscopy (AFM, NaniteAFM, Great Britain) in the 50 × 50 μm area. Surface roughness parameter S_a_, was determined using software that is an integral part of the device. As demonstrated by the conducted research, electrochemical anodization increases the roughness parameter S_a_ for both coatings that were produced at 5 V and 15 V. For the coating that was obtained at a voltage of 5V, the increase in the S_a_ parameter was almost threefold and for the coating obtained with 15V, more than five times. Also, the implantation of silver ions into electrochemically anodized coatings further increases the S_a_ parameter. In the case of the Ti6Al4V/TNT5/AgNPs coating, the S_a_ parameter increased by a further 32% and for the Ti6Al4V/TNT15/AgNPs coating by 9.3%. The implantation of silver ions into the Ti6Al4V substrate causes a threefold increase in surface roughness, from S_a_ = 0.027 µm to S_a_ = 0.078 µm (Figure 5).

#### 2.3.2. Hardness and Young’s Modulus

Hardness tests were carried out using a Berkovich indenter. All of the tested samples were subjected to 25 (5 × 5) measurements of nanoindentation. Individual indentations were spaced 20 μm apart (in both axes). Table 5 presents the mechanical properties measured in nanoindentation tests and Figure 6 exemplary hardness distribution on the area of 50 × 50 µm. The obtained results revealed that the surface implantation of the Ti6Al4V alloy with silver ions causes a slight increase in hardness, from 6.18 GPa to 6.81 GPa. On the other hand, after electrochemical anodization of the titanium alloy surface, the increase in hardness is greater than after surface implantation with silver ions. A particularly large, more than two and a half times, increase in hardness was noted for electrochemically anodized coatings that were obtained at 15 V (Ti6Al4V/TNT15).

For coating obtained at 5 V (Ti6Al4V/TNT5), the increase in hardness was not so significant; the value only increased by 20%. After implantation with silver ions electrochemically anodized coatings, it can be noticed that, depending on the anodizing voltage, the hardness either decreases or increases. In the case of anodized coating that was obtained at 5 V, after implantation with silver ions an increase in hardness by 33% (from 7.42 GPa to 9.46 GPa) is observed. However, for anodized coating that was obtained at 15 V, after implantation with silver ions, a 16% reduction in hardness is observed (from 16.23 GPa to 13.60 GPa). Similar changes after the silver ion implantation of anodized coatings, as in the case of hardness, can also be observed for the measured Young’s modulus, i.e., an increase in stiffness for the Ti6Al4V/TNT5/AgNPs composite and a decrease in stiffness for the Ti6Al4V/TNT15/AgNPs composite. In turn, the implantation with silver ions of the surface of the Ti6Al4V alloy results in the reduction of the Young’s modulus from 230.12 GPa to 187.54 GPa (18.5%).

#### 2.3.3. Adhesion Tests of Ti6Al4V/TNT and Ti6Al4V/TNT/AgNPs Composites

The coatings were subjected to five scratch tests (individual nanosporks were spaced apart by 250 μm). Table 6 presents aggregate results for all investigated coatings and Figure 7 shows exemplary curves that were obtained in the scratch test. As can be seen from the data presented in Table 6, electrochemical anodization at 15 volts allows for obtaining coatings with greater adhesion to the substrate than anodizing at five volts. The critical force resulting in the loss of adhesion is about 39% higher for the coating obtained at 15 volts than for the coating obtained at five volts.

In addition, the standard deviation of the average results of the critical force causing the loss of adhesion to the substrate is about three times greater in the case of coatings that were obtained with the voltage of five volts. Implantation of electrochemically anodized coatings with Ag ions contributes to changes in the critical force that causes loss of coating adhesion. Electrochemically anodized coatings that were obtained at 5 V, after their implantation with Ag ions, show an increase in critical force of about 39%, while the implantation with Ag ions of coatings obtained at 15 V causes a slight decrease in adhesion by about 3.6%.

### 2.4. Evaluation of Stability and Durability of Coating Materials in the Body Fluid Environment

The processes of silver ions releasing from the surface of Ti6Al4V/AgNPs and Ti6Al4V/TNT/AgNPs samples, immersed in phosphate-buffered saline (PBS) solutions at human body temperature (310 K), were studied for five weeks. The results of these investigations are presented in Figure 8.

Analysis of inductively coupled plasma mass spectrometry (ICP-MS) data revealed that 3 h after immersion of Ti6Al4V/AgNPs system in the PBS solution, the concentration of Ag^+^ ions was 0.18 ppm, and after 24 and 48 h, it was 0.40 and 0.86 ppm, respectively (Figure 8; extracted graph). Over the next 12 days, a further increase in the concentration of Ag^+^ in PBS solution was observed up to 2.52 ppm after 14 days.

During the next 14 days, the concentration of Ag^+^ ions did not change significantly remaining at 2.51–2.57 ppm (Figure 8). Studies of Ti6Al4V/TNT/AgNPs composites, which were produced by the deposition of silver nanoparticles on the surface of titanium dioxide nanotubes, showed that the Ag^+^ releasing processes were preceded in another way (Figure 8). In our experiments, we have used Ti6Al4V/TNT substrates of nanotube diameters ca. 35–45 nm (TNT5), 70–80 nm (TNT15), and 100–120 nm (TNT20). Obtained results revealed that during the first 14 days, the release of silver ions from the surface of all studied Ti6Al4V/TNT/AgNPs composites immersed in PBS solution, was not observed. After this time there was a slow increase of Ag^+^ concentration, reaching the highest value ca. 0.77 ppm (Ti6Al4V/TNT20/AgNPs) after 35 days, which was three times lower than in the case of the Ti6Al4V/AgNPs system (Figure 8). Simultaneously, the lowest concentration of Ag^+^, which amounted ca. 0.44 ppm (after 35 days) has been noticed for Ti6Al4V/TNT15/AgNPs. The obtained results revealed the clear impact of AgNPs diameter and the manner of their arrangement on the surface of TNT layers on the concentration of released silver ions (Figure 4).

## 3. Discussion

The implant samples that were fabricated by the selective laser sintering of Ti6Al4V powders (SLS technology) were used as substrates in all of our experiments, and the results are discussed in this paper. The appropriate porosity of substrates was obtained by covering their surface with TiO_2_ nanotube coatings (TNT), which were produced using the electrochemical anodization method [9,10,31]. In our works, we have focused on studies on amorphous Ti6Al4V/TNT systems that are produced using potentials: 5 V (Ti6Al4V/TNT5), 15 V (Ti6Al4V/TNT15), and 20 V (Ti6Al4V/TNT20), which showed the promising bioactivity [10]. The conversion of amorphous TiO_2_ layers into anatase phase during their heating up to 573 K were not noticed, which was confirmed by the analysis of IR and Raman spectra. This fact was important for our further works that are associated with the use of CVD technique in order to the enrichment of the Ti6Al4V and Ti6Al4V/TNT substrate surfaces with the AgNPs. Our earlier experience with the use of the different Ag(I) precursors in CVD experiments prompted us to choose Ag(O_2_CC_2_F_5_) as a suitable source of dispersed AgNPs [28,29]. However, during the recrystallization of this compound from a 1:2 EtOH/H_2_O mixture, the colorless crystals have been isolated after five days. The analysis of single crystals X-ray diffraction data proved the formation of trihydrate species of the general formula [Ag_5_(O_2_CC_2_F_5_)_5_(H_2_O)_3_] (Figure 1, Table 1). Three water molecules, which are presented in the structure of this compound (differently coordinated with silver atoms and taking part in possible interactions by hydrogen bonds with oxygen and fluorine atoms), should impact its properties as the CVD precursor. The results of volatility studies (VT IR and MS EI) showed that the heating of this compound in the range 303–503 K proceeded with its dehydration and the releasing of dimeric Ag(I) carboxylate species. Carried out studies exhibited that, in comparison to its anhydrous form, the isolated trihydrate crystals are characterized by the lower vaporization temperature at the pressure 5 × 10^−1^ hPa. Moreover, CVD experiments proved that the deposition of dispersed AgNPs proceeded with the lower deposition rate *r*_D_ = 2.25–2.57 mg·min^−1^ at *T*_D_ = 553 K (Table 3). The SEM images that are presented in Figure 4 indicated that spherical nanoparticles of silver of dimeters ca. 34–80 and 24–35 nm were localized on the surface of Ti6Al4V/TNT5/AgNPs and Ti6Al4V/TNT20/AgNPs coatings, respectively. Simultaneously, in the case of Ti6Al4V/TNT15/AgNPs, most of silver particles were localized on the walls, inside of tubes. The differences that are mentioned above can explain the noticed changes in wettability (hydrophilicity) and in the way of silver ions releasing from Ti6Al4V/TNT/AgNPs composites.

The direct consequence of the TNT layer formation on the surface of Ti6Al4V implant is the surface wettability increase with simultaneous surface free energy growth (Table 4). The obtained results are in good accordance with previous reports [32]. The enrichment of Ti6Al4V and Ti6Al4V/TNT surface by AgNPs lead to a decrease of their wettability and SFE value. The analysis of water contact angle changes revealed that the AgNPs deposition on the surface of hydrophilic surface of TNT coatings (water contact angle < 10 deg) lead to the increase of their hydrophobicity (water contact angle was changed to 110.2–124.2 deg) (Table 4). Simultaneously, it should be noted that hydrophobicity of studied samples decreases in the row: Ti6Al4V/TNT5/AgNPs > Ti6Al4V/TNT15/AgNPs > Ti6Al4V/TNT20/AgNPs. This effect can be related to the increase of nanotubes diameter from 35–45 nm up to 100–120 nm, and thus a higher ability to penetrate the interior of the nanotubes by the liquid. The increase of hydrophobicity of TNT layers (diameter 33 nm) after their decoration by AgNPs (diameter 35 nm) was also noticed by Caihong et al. [33]. The insertion of an AgNPs-enriched implant into an aqueous solution is associated with the oxidation of metal nanoparticles and the releasing of silver ions, which has direct impact on potential antimicrobial properties of the produced coatings [34]. Figure 8 shows that Ag^+^ ions were released with the high rate in the first 12 days from the surface of Ti6Al4V/AgNPs system. After this time, the release rate changes indicate saturated behavior, and the concentration of the Ag^+^ ions in PBS was close to 2.5 ppm. The higher concentration of these ions than 10 ppm in the human body can be toxic [35]. Zaho et al. showed a similar way of Ag^+^ releasing from the surface of Ti/AgNPs substrates, however the concentration of silver ions in PBS solution after seven days stabilized on the level 0.13 ppm [36]. In the case of Ti6Al4V/TNT/AgNPs coatings that are immersed in PBS solution, the different silver ion releasing pathway has been noticed (Figure 8). Independently to the TNT diameters (TNT5—35–45 nm, TNT15—70–80 nm, TNT20—100–120 nm), the silver ions release process was not observed in first seven days. After this time, the Ag^+^ ions concentration slowly increases, reaching values 0.44–0.77 ppm after 35 days dependently to the TNT diameter, AgNPs size, and the way of their distribution. Attention is drawn to the fact that the release rate of silver ions from Ti6Al4V/TNT20/AgNPs (lowest value of SFE and water contact angle 110.2 deg) is higher in comparison to Ti6Al4V/TNT15/AgNPs (highest value of SFE and water contact angle 124.2 deg) (Table 4). The earlier studies of Ti/TNT(anatase)/AgNPs composites (tube diameters were 50, 75, and 100 nm) revealed that Ag^+^ ions were releasing with the high rate in the first two days and maintaining concentration at the level 0.25–0.28 ppm [36]. Also, in this case, the highest release rate was noticed for AgNPs that were deposited on the surface of TNT layer consisted from tubes of diameter 100 nm. Our earlier studies of Ag^+^ ions releasing from Ti/TNT/AgNPs composites revealed that, after 21 days, the concentrations of these ions in PBS solution were close to 0.005–0.008 ppm, and after 28 days they increased to the level 0.15–0.22 ppm [9].

Up till now, the mechanical properties of TiO_2_ coatings that are produced by the electrochemical anodization of titanium alloys have been poorly explored. However, Young’s modulus, hardness, and adhesion of the coating to the substrate can be the decisive factors in terms of applications, for example, when considering the production of such coatings on implants elements. The strong integration of an implant with the bone tissue is crucial for the safe operation of the implant. The loss of coherence between the bone and the implant due to friction contributes to the implant wear. The abrasive wear of the implant may additionally cause inflammatory reactions in the patient’s body. Von Wilmowsky et.al reported in their work [37] that TiO_2_ nanotube coatings have shown a “good” qualitative adherence, but other reports [38,39] have qualified such coatings as “not very well adherent”. The results of our nano scratch-tests revealed that the adhesion of the Ti6Al4V/TNT15 coating to the substrate is slightly greater than the Ti6Al4V/TNT5 coating (Figure 7). This difference can be result in differences in the way of TNT coatings architecture. The TNT5 layer is composed of dense packed nanotubes of diameter ca. 35–45 nm and wall thicknesses ca. 12 nm, while the TNT15 consists of separated tubes of diameter ca. 70–80 nm and wall thicknesses ca. 20 nm. After the AgNPs deposition on the surface of TNT5 coating, the dispersed nanoparticles with diameters of 34–80 nm were located mainly on the layer surface, which should not impact the adhesion of the coating to the substrate. However, AgNPs of diameters that are similar to tube sizes can be located inside of tubes or close them. This can explain the adhesion increase of TNT5 coating from 197.7 mN up to 275.03 mN after the deposition AgNPs on its surface. In the case of TNT15 coating, which consists of tubes of diameters 70–80 nm (TNT15), the size of AgNPs decreased to 10–18 nm maximum. The metallic grains were located inside the tubes on their walls, which caused a slight decrease in the adhesion of the coating to the substrate (from 275.1 mN to 267.7 mN).

Analysis of data presented in Table 5 revealed a clear increase of nano-mechanical properties (hardness, Young’s modulus) of Ti6Al4V substrate surface after the formation Ti6Al4V/TNT system. However, it should be noted that the magnitude of these changes is associated with the morphology of the produced TNT coatings. The results of our measurements indicate that the hardness (7.42 GPa) and Young’s modulus (229.71 GPa) of Ti6Al4V/TNT5 coatings are significantly lower than that of Ti6Al4V/TNT15 coatings (hardness 16.23 GPa and Young modulus 350.64 GPa), despite the fact that they have a finer nanotubes structure. It can be explained by the increase of both stiffness and hardness of coatings, which results from an increase of the used voltage in the anodization process. The earlier works revealed that voltage increase (at a constant process time) is accompanied with an increase nanotubes diameter, wall thickness, and also their length [40]. Moreover, the increase in voltage is accompanied by the growth of barrier layer thickness in the lower part of the nanotube, which results in the formation of larger pores and greater distance between them [41]. As a result of these processes, the nanotubes of larger diameter and wall thickness are formed. Bauer et al. [42], revealed that the anodizing of titanium using voltages 10–25 V led to the formation of coatings that are composed of separated and ordered nanotubes, while the use of lower voltages led to the formation of a structure resembling an ordered network. For studied coatings, the above-mentioned morphology changes were noticed for TNT5 and TNT15 respectively. Simultaneously, the values of the surface roughness decreased from S_a_ = 0.137 µm for Ti6Al4V/TNT5 to S_a_ = 0.075 µm for Ti6Al4V/TNT15. The phase structure of nanotubes is the next important factor that can influence the hardness and stiffness of TNT coatings. Analysis of the previous reports showed that anodized TiO_2_ nanotubes are amorphous in nature [43,44,45]. However, the method and parameters of coatings production as well as their heat treatment may affect the occurrence of crystalline TiO_2_ phases, such as anatase, rutile, or a mixture of these polycrystalline forms. It was indicated that amorphous TiO_2_ nanotubes are softer than mixture of amorphous and crystallized TiO_2_ (anatase) nanotubes. Also, the geometry of nanotubes, i.e., their length and wall thickness will affect the hardness measurement result [46,47,48]. In addition, it is well-known that the radius of the nanoindenter’s tip rounding has an effect on the test results. The porosity of the coatings constitutes an additional parameter affecting their nano-mechanical properties. Munirathinam and Neelakantan reported [49] that porosity has a significant influence on the elastic modulus of the nanotubes. In our experiments, the potential of 15 V applied during the anodizing of the Ti6Al4V alloy was low enough for the formation of a structure with low porosity, as evidenced by the high Young’s modulus of the Ti6Al4V/TNT15 coating. After enriching produced coatings by AgNPs, the hardness of the Ti6Al4V/TNT5 coating increased from 7.42 GPa to 9.86 GPa, while for the Ti6Al4V/TNT15 coating, the hardness decreased from 16.23 GPa to 13.60 GPa. In both cases, the values of S_a_ parameters also increased (Figure 5). When considering that during the CVD process, the amorphousness of TNT coatings does not change, the increase of hardness and the roughness (from S_a_ = 0.137 µm up to S_a_ = 0.181 µm) of the Ti6Al4V/TNT5/AgNPs composite can be explained by the deposition of AgNPs on the surface of an ordered network, which consists of the densely packed TiO_2_ nanotubes. In the case of Ti6Al4V/TNT15/AgNPs, the increase of tubes diameter (70–80 nm) and the deposition of silver nanoparticles in their interior slightly increase the coating roughness (S_a_ = 0.082 µm), but its hardness decreases in comparison to the layer, which consists of separated and ordered TiO_2_ nanotubes.

## 4. Materials and Methods

### 4.1. Synthesis of Silver CVD Precursor and Conditions CVD Processes Carry Out

The silver(I) pentafluoropropionate has been synthesized according to previously reported procedure [28,29]. The slow recrystallization of this salt from the 1:2 EtOH/H_2_O solution led to the isolation of stable in air and colorless crystals after five days. The light sensitivity of these crystals required their storage in the light protected container.

Yield: 89.2%; anal. calculated for C_10_H_6_F_25_O_13_Ag_5_: C, 8.98%, H, 0.44%; Found: C, 9.09%, H, 0.47%. ^13^C NMR (75 MHz), δ (ppm): 59.04 (CF_2_), 117.49 (CF_3_), 167.25 (COO), ^19^F NMR (CDCl_3_, 376 MHz), δ (ppm): −82.91 (s, 3F), −118.38 (d, *J* = 21.1 Hz, 2F) Solid NMR spectra were recorded in a Varian Gemini 200 MHz NMR spectrometer in CDCl_3_ (Varian Inc., Palo Alto, CA, USA). Single crystal X-ray diffraction data were collected from a crystal of dimensions 0.57 × 0.51 × 0.38 nm with an Oxford Diffraction KM4 CCD diffractometer (Oxford Diffraction Ltd., Abingdon, Oxfordshire, UK) (Mo Kα about wavelength λ = 0.71073 Å). The structure was solved by direct methods and it was refined with the full-matrix least squares on F^2^ using the software package SHELX-97 [50]. All of the figures were prepared in DIAMOND [51] and ORTEP-3 [52]. CCDC: 1477237; contains the supplementary crystallographic data for [Ag_5_(O_2_CC_2_F_5_)_5_(H_2_O)_3_]. These data can be obtained free of charge via http://www.ccdc.cam.ac.uk/conts/retrieving.html, or from the Cambridge Crystallographic Data Centre, 12 Union Road, Cambridge CB2 1EZ, UK; fax: (+44)1223-336-033; or e-mail: deposit@ccdc.cam.ac.uk. Crystal data and structure refinement for this compound are given in Table 7.

The enrichment of AgNPs on the surface of Ti6Al4V and Ti6Al4V/TNT substrates were carried out while using CVD method (the horizontal hot-wall reactor) under the conditions that are presented in Table 2 [29].

### 4.2. The Production of Ti6Al4V/TNT Substrates and Its Characteristics

Titanium dioxide nanotube layers (TNTs) were fabricated on the surface of implants of the radial bone (total area implant—20.53 cm^2^, formed by selective laser sintering (SLS) of Ti6Al4V powder) as a result of the electrochemical anodic oxidation, according to the method previously described [9,10,31]. This process was carried out at the following voltages: 5 V (TNT5), 15 V (TNT15), and 20 V (TNT20). The anodizing time was t = 30 min. The morphology of the produced coatings was examined using a Quanta scanning electron microscope with field emission (SEM, Quanta 3D FEG, Huston USA). The structure of the produced TiO_2_ nanotube layers was studied while using Raman spectroscopy (RamanMicro 200 PerkinElmer (PerkinElmer Inc., Waltham, MA, USA) (λ = 785 nm)) and diffuse reflectance infrared Fourier transform spectroscopy (DRIFT, Spectrum2000, PerkinElmer Inc., Waltham, MA, USA).

### 4.3. Measurement of the Contact Angle and Surface Free Energy of Biomaterials

Determination of the wettability was carried out by the measuring of the contact angle. The contact angle was measured using a goniometer with drop shape analysis software (DSA 10 Krüss GmbH, Hamburg, Germany). The liquids that were selected for measuring the contact angle were distilled water (H_2_O) and diiodomethane (CH_2_I_2_). In the case of distilled water, the volume of the drop in the contact angle measurement was 3 μL, and in the case of diiodomethane 4 μL. The measurement of the contact angle was carried out immediately after deposition of the drop. In order to determine the surface free energy, mathematical calculations were performed using the Owens-Wendt method. Each measurement was carried out three times.

### 4.4. Topographies and Mechanical Properties of the Produced Nanocoatings on the Surface of 3D Printed Implants

Surface topographies were examined by means of atomic force microscopy (AFM, NaniteAFM, Nanosurf AG, Liestal, Switzerland) using a contactless module with a force of 55 mN in the 50 × 50 μm area. Hardness tests and Young modulus measurements were carried out using a nanoindenter (NanoTest Vantage, Micro Materials Ltd., Wrexham, UK) using a pyramidal, diamond, three-sided Berkovich indenter, with an apical angle of 124.4°. Hardness tests were performed for the loads of 10 mN. The time of load increase from the zero value to the maximum load 10 mN was 15 s. Indentation involving one cycle with 5 s dwell at maximum load. Hardness values (H), reduced Young’s modulus (Er), and Young’s modulus were determined using the Oliver-Pharr method using the NanoTest results analysis program. In order to convert the reduced Young’s modulus into the Young’s modulus, a Poisson coefficient of 0.25 was assumed for the coatings.

Tests of coatings adhesion were made using nanoindenter (NanoTest Vantage, Micro Materials Ltd., Wrexham, UK) and using the Berkovich indenter, as in the case of the nanoindentation tests.

The parameters of scratch tests were as follows: scratch load—0 to 500 mN, loading rate—3.3 mN/s, scan velocity—3 μm/s, and scan length—500 μm. Based on the dependence of the friction force (Ft) on the normal force (Fn) in the program for the analysis of NanoTest results, the values of critical friction force (Lf) and critical force (Lc), which caused the separation of the layer from the substrate, were determined.

### 4.5. Evaluation of Stability and Durability of Coating Materials in the Body Fluid Environment

The analysis was carried out on Ag-enriched (a) titanium foil (Ti6Al4V, gradation 5, 99.7% purity, STREM) and (b) titanium foil with tiatnium dioxide nanotube modified surface (i.e., arrangements Ti/Ag and Ti/TNT5/Ag; Ti/TNT15/Ag; Ti/TNT20/Ag). Both variants were cut into 7 mm × 7 mm pieces. These composites were additionally protected with polyglycolide (PGA) and were also analyzed. The prepared materials were immersed in 15 mL of buffered saline solution, with the concentration of ions and the osmotic pressure being comparable to that which prevails in human body fluids. This solution was made by dissolving a PBS tablet with the following composition: 140 mM NaCl, 10 mM phosphate buffer, 3 mM KCl in 100 mL distilled water. The samples were kept in an incubator at 310 K for 1, 2, 3, 4, 6, 7, 9, 10, 13, 14, 21, 26, 28, and 35 days. Estimation of silver concentration was performed by mass spectrometry with plasma ionization inductively coupled to a quadrupole analyzer using an ICP-MS 7500 CX spectrometer with Agilent Technologies collision chamber (Agilent Technologies Inc., Tokyo, Japan).

## 5. Conclusions

The direct result of our works was the fabrication of coatings that are composed of the dispersed AgNPs and/or the TNT/AgNPs nanocomposites on the surface of Ti6Al4V implants, produced in SLS technology. The TNT layers were produced using electrochemical anodization of Ti6Al4V at 5, 15, and 20 V, while the CVD method was used in order to enrich Ti6Al4V and Ti6Al4/TNT substrates by silver nanoparticles. The carried out CVD experiments proved the utility of [Ag_5_(O_2_CC_2_F_5_)_5_(H_2_O)_3_)] as a precursor of metallic silver nanoparticles. The enrichment of Ti6Al4V and Ti6Al4V/TNT substrates with AgNPs increased their surface free energy, roughness, hardness, and Young’s modulus. It also caused the increase of the hydrophobic properties. Studies of Ti6Al4V/TNT/AgNPs composites that were immersed in PBS solution proved that the concentration of silver ions released form the surface of these materials changes between 0.44 and 0.77 ppm after 35 days. This value is definitely below the critical level, which could have any negative effect on mammalian cells [35].

## Figures and Tables

**Figure 1 ijms-19-03962-f001:**
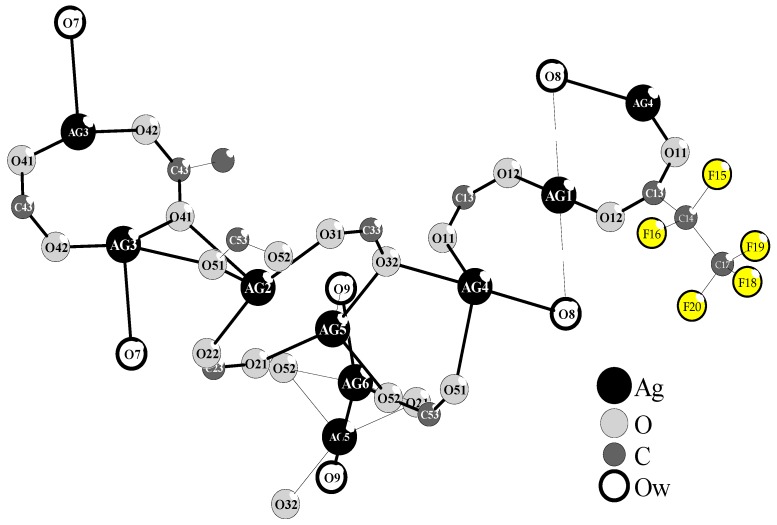
The scheme of the structure of [Ag_5_(O_2_CC_2_F_5_)_5_(H_2_O)_3_]. For clarity, the position of only one C_2_F_5_ group has been presented (the full structure image is shown in the Appendix A: pp11a-2000-sch_shape-13b3-checkcif).

**Figure 2 ijms-19-03962-f002:**
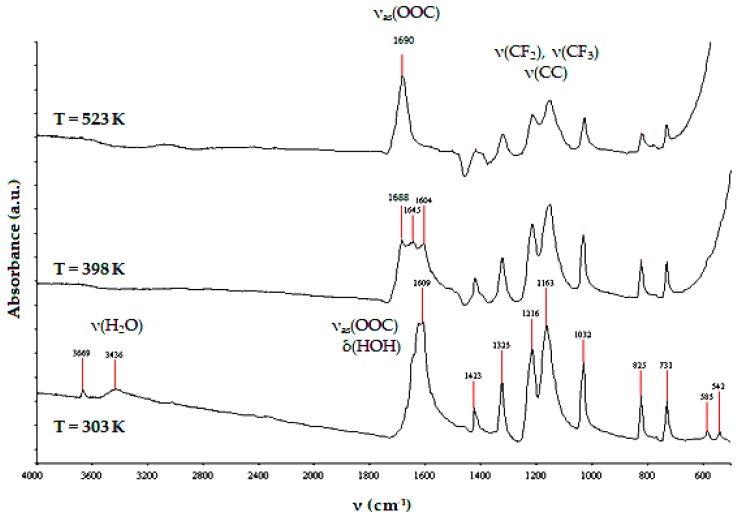
IR spectra of [Ag_5_(O_2_CC_2_F_5_)_5_(H_2_O)_3_] registered at 303, 398, and 523 K.

**Figure 3 ijms-19-03962-f003:**
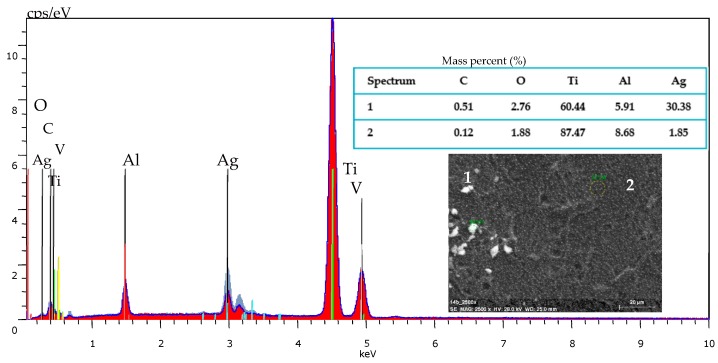
Energy Dispersive Spectroscopy (EDS) spectrum of Ti6Al4V/AgNPs sample and quantitative data.

**Figure 4 ijms-19-03962-f004:**
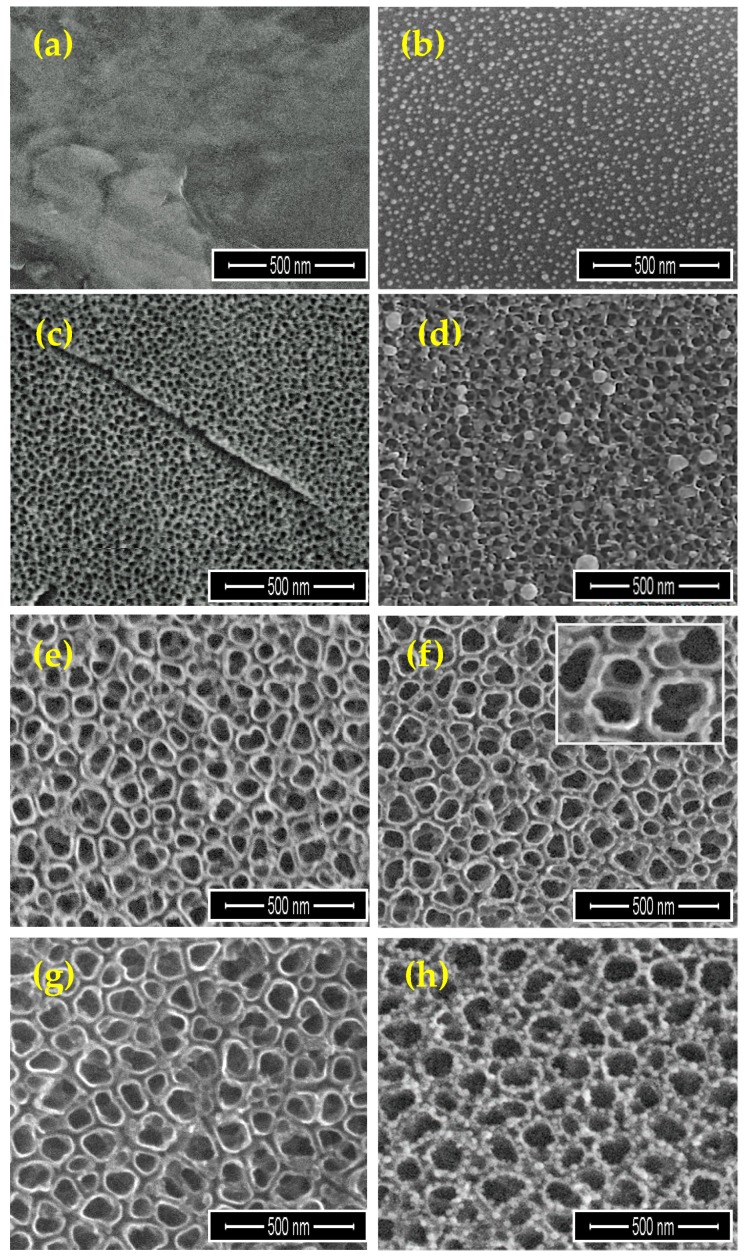
Scanning electron microscopy (SEM) images of Ti6Al4V (**a**), Ti6Al4V/AgNPs (**b**), Ti6Al4V/TNT5 (**c**), Ti6Al4V/TNT5/AgNPs (**d**), Ti6Al4V/TNT15 (**e**), Ti6Al4V/TNT15/AgNPs (**f**), Ti6Al4V/TNT20 (**g**), and Ti6Al4V/TNT20/AgNPs (**h**).

**Figure 5 ijms-19-03962-f005:**
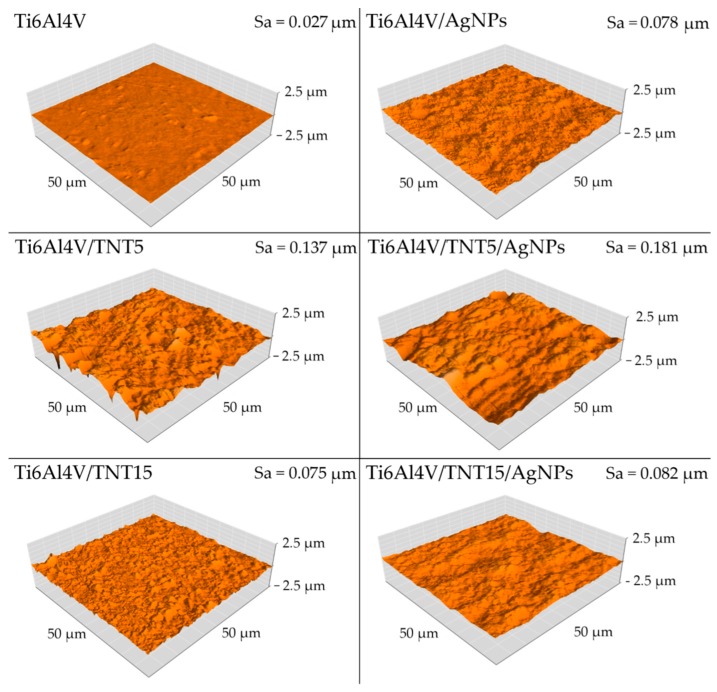
Atomic force microscopy (AFM) surface topography and S_a_ parameter of reference Ti6Al4V, Ti6Al4V/AgNPs, and Ti6Al4V/TNT/AgNPs composites.

**Figure 6 ijms-19-03962-f006:**
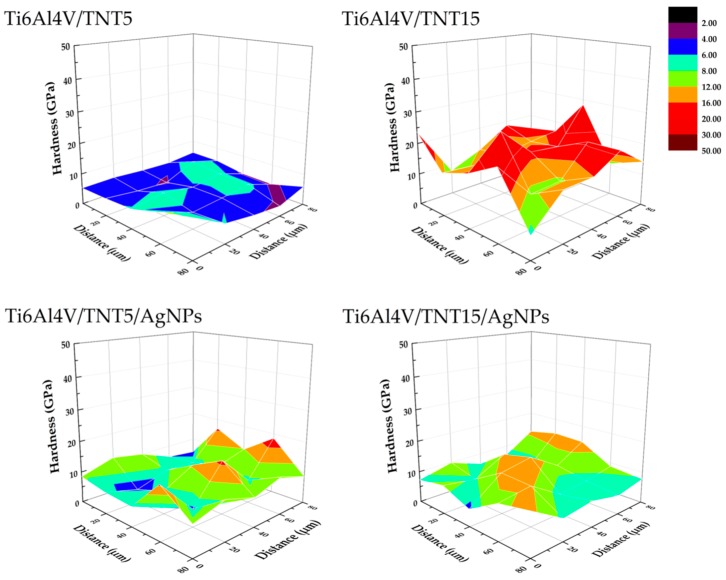
Hardness distribution of Ti6Al4V/TNT5, Ti6Al4V/TNT15, Ti6Al4V/TNT5/AgNPs, and Ti6Al4V/TNT15/AgNPs composites.

**Figure 7 ijms-19-03962-f007:**
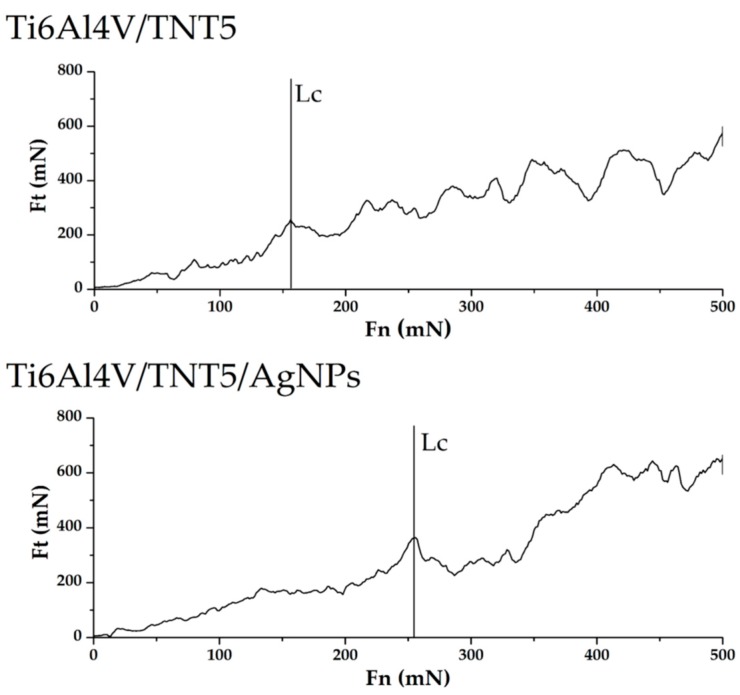
Examples of results obtained in the scratch test for Ti6Al4V/TNT5 coating and for Ti6Al4V/TNT5/AgNPs composite.

**Figure 8 ijms-19-03962-f008:**
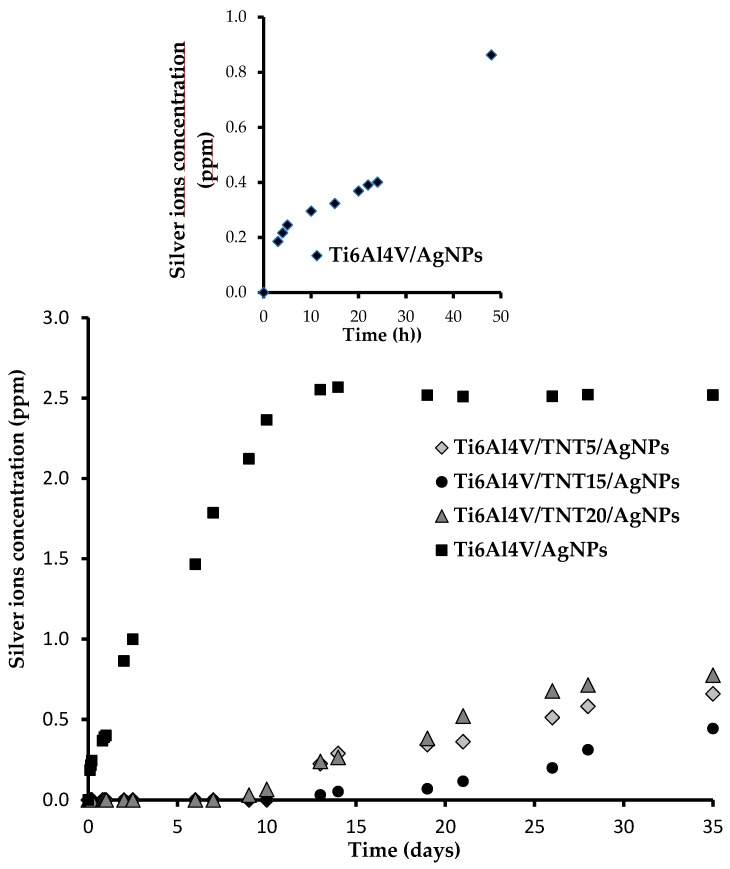
Silver ions released from Ti6Al4V/AgNPs coatings and Ti6Al4V/TNT5/AgNPs; Ti6Al4V/TNT15/AgNPs; and, Ti6Al4V/TNT20/AgNPs coatings immersed in phosphate buffered saline (PBS). The extracted graph shows the concentration changes of silver ions released from the surface of Ti6Al4V/AgNPs in the first 48 h after immersion off the sample in PBS solution.

**Table 1 ijms-19-03962-t001:** Selected bonds lengths [Å] and angles [°] for [Ag_5_(O_2_CC_2_F_5_)_5_(H_2_O)_3_].

**Bond Length**					
Ag1-O12	2.138(5)	Ag3-O42	2.205(4)	Ag5-O21	2.234(5)
Ag1-O12 ^i^	2.138(5)	Ag3-O41	2.218(4)	Ag5-O32	2.249(4)
Ag1-O8	2.825(6)	Ag3-O7	2.547(4)	Ag5-O52	2.379(5)
Ag1-Ag4 ^i^	3.0058(5)	Ag3-O51	2.588(4)	Ag5-O9	2.641(6)
Ag1-Ag4	3.0058(5)	Ag3-Ag3 ^iii^	2.8932(9)	Ag5-Ag2 ^ii^	2.8951(7)
Ag2-O31 ^ii^	2.219(5)	Ag4-O11	2.324(4)	Ag5-Ag6	3.239(2)
Ag2-O22	2.237(5)	Ag4-O7	2.426(4)	Ag6-O9	2.318(7)
Ag2-O51	2.553(4)	Ag4-O51	2.511(4)	Ag6-O9 ^iv^	2.412(8)
Ag2-O41	2.610(4)	Ag4-O32	2.540(5)	Ag6-O52 ^iv^	2.472(6)
Ag2-Ag5 ^ii^	2.8950(7)	Ag4-O8	2.577(6)	Ag6-O52	2.579(6)
Ag2-Ag5	3.3236(8)			Ag6-O21	2.593(5)
**Angles**					
O12-Ag1-O12 ^i^	180.0	O42-Ag3-O41	162.69(16)	O21-Ag5-O32	155.4(2)
O12-Ag1-Ag4 ^i^	95.09(12)	O42-Ag3-O7	91.16(14)	O21-Ag5-O52	93.4(2)
O12 ^i^ -Ag1-Ag4 ^i^	84.91(12)	O41-Ag3-O7	104.90(15)	O32-Ag5-O52	108.9(2)
O12-Ag1-Ag4	84.91(12)	O42-Ag3-O51	106.76(16)	O21-Ag5-Ag2 ^ii^	82.41(13)
O12 ^i^ -Ag1-Ag4	95.09(12)	O41-Ag3-O51	84.01(16)	O32-Ag5-Ag2 ^ii^	81.11(12)
Ag4 ^i^ -Ag1-Ag4	180.0	O7-Ag3-O51	75.23(13)	O52-Ag5-Ag2 ^ii^	156.0(2)
O31 ^i^ -Ag2-O22	159.63(18)	O42-Ag3-Ag3 ^iii^	82.99(11)	O21-Ag5-Ag6	52.75(13)
O31 ^i^ -Ag2-O51	91.49(16)	O41-Ag3-Ag3 ^iii^	79.70(11)	O32-Ag5-Ag6	136.00(13)
O22-Ag2-O51	108.03(16)	O7-Ag3-Ag3 ^iii^	158.57(10)	O52-Ag5-Ag6	51.94(14)
O31 ^i^ -Ag2-Ag5 ^ii^	81.38(12)	O51-Ag3-Ag3 ^iii^	126.20(10)	Ag2 ^ii^-Ag5-Ag6	133.99(5)
O22-Ag2-Ag5 ^ii^	78.26(13)	O11-Ag4-O7	153.99(15)	O21-Ag5-Ag2	134.02(15)
O51-Ag2-Ag5 ^ii^	163.39(10)	O11-Ag4-O51	126.55(15)	O32-Ag5-Ag2	61.71(13)
O31 ^ii^ -Ag2-Ag5	61.33(14)	O7-Ag4-O51	78.78(14)	O52-Ag5-Ag2	82.47(13)
O22-Ag2-Ag5	120.51(13)	O11-Ag4-O32	88.59(16)	Ag2 ^ii^-Ag5-Ag2	83.578(19)
O51-Ag2-Ag5	67.06(10)	O7-Ag4-O32	100.33(14)	Ag6-Ag5-Ag2	133.31(6)
Ag5 ^ii^ -Ag2-Ag5	96.42(2)	O51-Ag4-O32	85.85(15)	O9-Ag6-O9 ^iv^	154.41(13)
		O11-Ag4-O8	93.10(17)	O9-Ag6-O52 ^iv^	98.8(3)
		O7-Ag4-O8	78.17(17)	O9 ^iv^-Ag6-O52 ^iv^	78.7(2)
		O51-Ag4-O8	93.07(17)	O9-Ag6-O52	78.3(2)
		O32-Ag4-O8	178.31(16)	O9 ^iv^-Ag6-O52	93.6(2)
		O11-Ag4-Ag1	74.81(10)	O52 ^iv^-Ag6-O52	156.09(11)
		O7-Ag4-Ag1	79.64(9)	O9-Ag6-O21	79.1(2)
		O51-Ag4-Ag1	148.61(10)	O9 ^iv^-Ag6-O21	124.0(2)
		O32-Ag4-Ag1	120.40(11)	O52 ^iv^-Ag6-O21	122.1(2)
		O8-Ag4-Ag1	60.22(15)	O52-Ag6-O21	81.03(19)
				O9-Ag6-Ag5	53.70(16)
				O9 ^iv^-Ag6-Ag5	134.01(17)
				O52 ^iv^-Ag6-Ag5	147.11(18)
				O52-Ag6-Ag5	46.58(12)
				O21-Ag6-Ag5	43.30(11)

^i^ -x,-y,-z; ^ii^ -x,-y-1,-z-1; ^iii^ -x,-y,-z-1; ^iv^ -x-1,-y-1,-z-1.

**Table 2 ijms-19-03962-t002:** Silver(I) fragmentation ions present on the mass spectra (MS EI) of [Ag_5_(O_2_CC_2_F_5_)_5_(H_2_O)_3_] registered at 403 and 513 K.

Fragmentation Ions	*m*/*z*	403 K	503 K	523 K
[Ag(CO)]^+^	136	8	-	-
[Ag(O_2_C)]^+^	147	21	11	4
[Ag(O_2_CF)]^+^	171	23	>2	-
[Ag(C_2_F_5_)]^+^	209	10	31	12
[Ag(O_2_CC_2_F_5_)(H_2_O)]^+^	289	100	-	-
[Ag_2_(C_2_F_5_)]^+^	335	58	100	38
[Ag_2_(O_2_CC_2_F_5_)]^+^	379	-	68	6
[Ag(O_2_CC_2_F_5_)_2_(H_2_O)]^+^	452	10	-	-
[Ag_2_(O_2_CC_2_F_5_)(C_2_F_5_)]^+^	498	30	5	>2
[Ag_2_(O_2_CC_2_F_5_)_2_(CO)]^+^	586	>2	>1	-
[Ag_2_(O_2_CC_2_F_5_)_2_(CO)_2_]^+^	598	>1	>1	-
[Ag_3_(O_2_CC_2_F_5_)(C_2_F_5_)(CO)]^+^	635	>2	-	-
[Ag_3_(O_2_CC_2_F_5_)_2_(CO)]^+^	679	>1	-	-
[Ag_2_(O_2_CC_2_F_5_)_3_(OC)(H_2_O)]^+^	752	>1	-	-
[Ag_2_(O_2_CC_2_F_5_)_3_(OOC)(H_2_O)_2_]^+^	784	>1	-	-

**Table 3 ijms-19-03962-t003:** Summary of chemical vapor deposition (CVD) conditions for the deposition of silver nanograins.

	[Ag_5_(O_2_CC_2_F_5_)_5_(H_2_O)_3_]	Ag(O_2_CC_2_F_5_) [29]
Total reactor pressure (p) [hPa]	5 × 10^−1^	4
Substrate temperature (T_D_) [K]	553	563
Vaporization temperature (T_V_) [K]	508	513
Deposition rate (*r*_D_) [mg·min^−1^]	2.25–2.57	2.56
Carrier gas	Ar	Ar
Deposition time [min]	30	30
Precursor mass [mg]	100	100

**Table 4 ijms-19-03962-t004:** The results of the contact angle measurement, which was made three times using distilled water and diiodomethane and the results of the surface free energy (SFE) of biomaterials used in Owens-Wendt method.

Biomaterial Sample	Average Contact Angle [°] ± Standard Deviation		SFE [mJ/m^2^]
	Measuring Liquid		
	Water	Diiodomethane	
Ti6Al4V	108.3 ± 0.09	37 ± 0.16	45.37 ± 0.05
Ti6Al4V/AgNPs	131.9 ± 0.12	89.6 ± 0.50	15.09 ± 0.09
Ti6Al4V/TNT5	˂10	36 ± 6.82	˃72.06
Ti6Al4V/TNT15	˂10	32.3 ± 2.75	˃72.30
Ti6Al4V/TNT20	˂10	30.7 ± 2.18	>72.42
Ti6Al4V/TNT5/AgNPs	124.2 ± 0.06	41.9 ± 0.47	51.97 ± 0.15
Ti6Al4V/TNT15/AgNPs	120.5 ± 0.1	67.3 ± 0.96	28.46 ± 0.23
Ti6Al4V/TNT20/AgNPs	110.2 ± 0.5	72.3 ± 0.73	21.7 ± 0.05

**Table 5 ijms-19-03962-t005:** Mechanical properties (hardness, Young’s Modulus and maximum depth of indentation) of reference Ti6Al4V, Ti6Al4V/AgNPs, and Ti6Al4V/TNT/AgNPs composites.

Biomaterial Sample	Hardness [GPa]	Young’s Modulus [GPa]	Maximum Depth of Indentation [nm]
Ti6Al4V	6.18 ± 2.88	230.12 ± 21.68	162.18 ± 12.18
Ti6Al4V/AgNPs	6.81 ± 2.55	187.54 ± 54.33	253.09 ± 51.55
Ti6Al4V/TNT5	7.42 ± 3.30	229.71 ± 88.07	302.40 ± 61.85
Ti6Al4V/TNT15	16.23 ± 8.81	350.64 ± 157.57	168.11 ± 46.04
Ti6Al4V/TNT5/AgNPs	9.86 ± 4.61	253.93 ± 87.14	211.53 ± 56.38
Ti6Al4V/TNT15/AgNPs	13.60 ± 7.24	287.03 ± 92.92	184.46 ± 40.60

**Table 6 ijms-19-03962-t006:** Results of nano scratch-tests of Ti6Al4V/TNT and Ti6Al4V/TNT/Ag composites.

	Nano Scratch—Test Properties	
Biomaterial Sample	Critical Friction [mN]	Critical Load [mN]
Ti6Al4V/TNT5	155.76 ± 69.02	197.713 ± 78.62
Ti6Al4V/TNT15	234.68 ± 21.05	275.03 ± 28.91
Ti6Al4V/TNT5/AgNPs	213.57 ± 49.50	275.11 ± 58.15
Ti6Al4V/TNT15/AgNPs	238.27 ± 53.54	267.74 ± 75.73

**Table 7 ijms-19-03962-t007:** Crystal data and structure refinement for [Ag_5_(O_2_CC_2_F_5_)_5_(H_2_O)_3_)].

Formula sum	C_15_ H_6_ Ag_5_ F_25_ O_13_
Formula weight	1408.55
Crystal system	triclinic
Space group	P-1
Unit cell dimensions	*a* = 11.3277(5) Å *b* = 13.0765(5) Å *c* = 13.7547(5) Å *α* = 116.746(4)° *β* = 100.869(3)° *γ* = 99.819(3)°
Cell volume [Å^3^]	1709.36(13)
Density (calculated) [Mg/m^3^]	2.737
Z	2
Absorption coefficient [mm^−1^]	3.005
F(000)	1320
Crystal size [mm]	0.57 × 0.51 × 0.38
Theta range for data collection [deg^o^]	2.16 to 26.37
Index ranges	−14 ≤ h ≤ 14 −16 ≤ k ≤ 16 −17 ≤ l ≤ 17
Reflections collected	18624
Reflections unique/R_int_	6960/0.0424
Completeness to theta = 26.37	99.5%
Transmission Max/Min	0.3947/0.2792
Refinement method	Full-matrix least-squares on F^2
Data/restraints/parameters	6960/20/592
Goodness-of-fit on F^2	1.043
Final R indices [I > 2sigma(I)]	R1 = 0.0474 wR2 = 0.1339
R indices (all data)	R1 = 0.0627 wR2 = 0.1439
Largest diff. peak and hole [e.Å^−3^]	0.937 and −0.846

R1 = Σ||F_o_ | − |F_c_||/Σ|F_o__|_; wR2 = {Σ[w(F_o_^2^ − F_c_^2^)^2^]/Σ[w(F_o_^2^)^2^]}^1/2^.

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
