# Peer review of "Studies on Silver Ions Releasing Processes and Mechanical Properties of Surface-Modified Titanium Alloy Implants"

_ijms, 2018, doi:10.3390/ijms19123962_

Reviewer 1 Report

The manuscript is well-written and pleasant to read. 

Several suggestions: 

1)      The abstract does not clearly state the unique findings from this work.

2)      Can the authors state the positions of the C2F5 groups in Fig. 1.

3)      Have the authors conducted quantitative analysis on the EDS data to determine the relative chemical compositions of the AgNPs  on the surface of Ti6Al4V implant (Fig. 3)?

4)      How is the surface topography information important in this study? This should be clearly stated in the Introduction section  as well as the Discussions section.

5)      The authors conclude saying that ‘this result indicates that the nanocomposites produced --- should be not toxic to the cells  of the human body.’ However, the authors have not provided scientific evidence for this claim.

Author Response

Thank you for your review and valuable remarks and comments. I enclose the answers to them.

1) The abstract does not clearly state the unique findings from this work.

In the new version of Manuscript the reviewer suggestion has been taken into account and the abstract has been changed in order to clearly state the unique findings of our work.

2) Can the authors state the positions of the C2F5 groups in Fig. 1.

For clarity Figure 1 has been changed. The position of one of the C2F5 groups has been marked on the structure of Ag(I) complex.

3) Have the authors conducted quantitative analysis on the EDS data to determine the relative chemical compositions of the AgNPs on the surface of Ti6Al4V implant (Fig. 3)?

According to the reviewer suggestion EDS quantitative data have been added to the Figure 3.

4) How is the surface topography information important in this study? This should be clearly stated in the Introduction section as well as the Discussions section.

The surface topography measurement has been added to this studies, as we wanted to obtain the full characterization of the materials. In fact we are preparing now the next manuscript, in which we point out the influence of the surface roughness on the fibroblasts and osteoblasts adhesion and proliferation, as well as on the bacterial biofilm formation. However we have added to the Discussion section additional sentences about the directions of surface roughness changes during the enrichment with silver nanoparticles.

5) The authors conclude saying that ‘this result indicates that the nanocomposites produced --- should be not toxic to the cells of the human body.’ However, the authors have not provided scientific evidence for this claim.

According to the reviewer remark this sentence has been modified and adequate reference has been added.

Reviewer 2 Report

Dear authors,

The manuscript entitled „Studies on Silver Ions Releasing Processes and Mechanical Properties of Surface-modified Titanium  Alloy Implants” authored by Aleksandra Radtke, Michalina Ehlert, Marlena Grodzicka, Tadeusz M. Muzioł, Marek Szkodo, Michał Bartmański, and Piotr Piszczek contains some interesting results for IJMS readers but few explanations are missing. After analyzing the manuscript I have few suggestions and questions for the authors listed below.

 1.      The information from lines 106-108 are repeated in the lines 115-117. Let just one pargraph that combines the information.

2.      Table 1. Merge the cells from the first row of the table and also the row with Angles.

3.      Line 124: Should be deleted the word „the” before heating.

4.      Line 130: letter missing from the parantheses, „an” should be „and”.

5.      Figure 2: I suggest to change the name of Y axis in „Absorbance (a.u.)”.

6.      Lines 139-140: A reference should be given for the species identification by MS EI.

7.      Figure 3: Should be used a larger font size on the axis and on the figure notations.

8.      The resolution of Figure 4 should be improved and maybe use yellow font color for a better visibility of notations.

9.      Line 188: The abbreviation of surface free energy is SFE not SEP. This should be corrected in the whole manuscript.

10.  Table 4: Use dot instead of comma. Explaination should be given why the water contact angle decreases for the sample Ti6Al4V/TNT20/AgNPs when compare to Ti6Al4V/TNT5/AgNPs, Ti6Al4V/TNT15/AgNPs and Ti6Al4V/AgNPs.

11.  After what time was recorded the contact angle when the drop was deposited on the substrate??

12.  I recommend that the release study should be presented after the mechanical properties evaluation for a better overview and discussion of the results.

13.  Line 469: Give the temperature in Interational System of Units.

Author Response

Thank you for your review and valuable remarks and comments. I enclose the answers to them.

1.      The information from lines 106-108 are repeated in the lines 115-117. Let just one pargraph that combines the information.

The sentence in the lines 115-117 has been removed.

2.      Table 1. Merge the cells from the first row of the table and also the row with Angles.

The Table 1 has been corrected

3.      Line 124: Should be deleted the word „the” before heating.

The error has been corrected.

4.      Line 130: letter missing from the parantheses, „an” should be „and”.

The error has been corrected.

5.      Figure 2: I suggest to change the name of Y axis in „Absorbance (a.u.)”.

According to the reviewer suggestion, the description of the Y axis has been changed.

6.      Lines 139-140: A reference should be given for the species identification by MS EI.

The reference data have been added.

7.      Figure 3: Should be used a larger font size on the axis and on the figure notations.

 According to the reviewer suggestion Figure 3 was changed.

8.      The resolution of Figure 4 should be improved and maybe use yellow font color for a better visibility of notations.

 Figure 4 was changed.

9.      Line 188: The abbreviation of surface free energy is SFE not SEP. This should be corrected in the whole manuscript.

The error has been corrected.

10.  Table 4: Use dot instead of comma. Explaination should be given why the water contact angle decreases for the sample Ti6Al4V/TNT20/AgNPs when compare to Ti6Al4V/TNT5/AgNPs, Ti6Al4V/TNT15/AgNPs and Ti6Al4V/AgNPs.

Data in Table 4 have been corrected.  Thank you for your attention. Unfortunately, in Table 4 I made a mistake in entering the data on hydrophilicity of Ti6Al4V/TNT5/AgNPs and Ti6Al4V/TNT15/AgNPS coatings. In the corrected version of manuscript this error have been corrected, and the  explanation of the noticed hydrophilicity change are presented in lines: 364-366: ”This effect can be related to the increase of nanotubes diameter from 35-45 nm up to 100-120 nm, and thus a higher ability to penetrate the interior of the nanotubes by the liquid.”

11.  After what time was recorded the contact angle when the drop was deposited on the substrate??

The measurement of the contact angle was carried out immediately after deposition of the drop.

12.  I recommend that the release study should be presented after the mechanical properties evaluation for a better overview and discussion of the results.

According to the reviewer remark, we have change the order of points 2.3 and 2.4.

13.  Line 469: Give the temperature in Interational System of Units.

The temperature was saved in SI units.